# High Temperature Friction and Wear Behavior of PTFE/MoS$_2$ Composites

**Jinming Zhen** [1,*], **Yunxiang Han** [1], **Huabao Wang** [2], **Zhenguo Jiang** [2], **Li Wang** [3], **Yuqiang Huang** [4], **Zhengfeng Jia** [1] and **Ran Zhang** [1]

1 College of Materials Science and Engineering, Liaocheng University, Liaocheng 252059, China
2 Shandong Dehai Youli New Energy Co., Ltd., Liaocheng 252059, China
3 Shandong Litong Clutch Manufacturing Co., Ltd., Liaocheng 252059, China
4 Yantai Evergrande Polymer Co., Ltd., Yantai 264006, China
* Correspondence: zhenjinming@lcu.edu.cn; Tel.: +86-135-6304-0259

**Abstract:** High performance polymer matrix composites with low friction and wear rate are of urgent requirement in sliding bearings and gaskets. In this study, the PTFE/MoS$_2$ composites were prepared and the effect of testing temperature on the tribological properties were investigated. Results show that the friction coefficient and wear rate are approximately (0.14–0.19) and (4.18–13.38 $\times$ 10$^{-4}$ mm$^3$/Nm) at testing temperatures from 25 to 250 °C, respectively. At testing temperatures above 200 °C, the coefficient of friction of the composite with the addition of MoS$_2$ is lower than that of pure PTFE, while the wear rate of the composite material with the addition of 2 wt.% and 5 wt.% MoS$_2$ is lower than that of pure PTFE at temperatures above 150 °C. At low testing temperatures (25–100 °C), the main wear mechanism is that of slight abrasive wear, while from 150 °C to 250 °C, the main wear mechanism transformed to fatigue and severe abrasive wear.

**Keywords:** PTFE matrix composite; dry sliding; MoS$_2$; high temperature





## 1. Introduction

Polytetrafluoroethylene (PTFE) is one of the remarkable materials that has very promising application prospects in mechanical industry due to its excellent thermal stability, low friction and chemical resistance properties [1–6]. However, PTFE has a poor wear resistance, and high wear will occur in the tribopaired materials while starting and stopping devices during the service process, so high-performance composites with excellent self-lubricating and wear resistance properties are urgently required to increase the service life.

Pure polymer composites have the disadvantages of low hardness, poor thermal conductivity, easy to creep/entangle, poor processing performance, etc. Therefore, they need to be modified to increase the mechanical and frictional properties. In terms of filling modification, researchers have conducted a lot of exploratory works, wherein they are mostly filled and modified with organic particles, inorganic particles, polymer, etc. [7–11]. Krick et al. [12] demonstrate that the addition of iron–cobalt (FeCo) microparticles could increase the wear resistance of PTFE composites due to their tribochemical mechanisms as compared to the elemental Fe- or Co-filled composites. Xie et al. [13] prepared several PTFE composites with Polyetheretherketone (PEEK), WS$_2$, and Si$_3$N$_4$, and the results showed that the number of layers of the transfer film of the PEEK/PTFE composite was the largest (four layers) and that of the WS$_2$/PTFE composite was the smallest (one layer). Wang et al. [14] enhanced the UHMWPE composite using a hybrid of attapulgite nanofibers with titanium dioxide nanoparticles and investigated the frictional behavior; the results showed that the network structure formed by the fillers improved the thermal stability of the composites and the optimal wear resistance was obtained at 5 wt.% filler content. Xie et al. [15] reported that the addition of LaF$_3$/CeF$_3$ can decrease transfer film thickness and promote a thin and uniform film formed on the worn surface than the single addition

of LaF$_3$ or CeF$_3$. Sawae et al. [16] prepared single-filler PTFE composites with different filler materials and evaluated the tribological behavior in gaseous hydrogen; results showed that the carbon fiber-filled PTFE indicates excellent low friction and low wear in hydrogen. Jin et al. [17] studied the lamellar sodium-montmorillonite (NaMMT) as a reinforcing filler to improve the abrasion resistance of polyethersulfone (PES)/PTFE-based polymers and found that the sample with 20 wt.% NaMMT exhibited the best anti-wear performance. Moreover, the additive can also promote the formation of lubricating layer on the worn surface. Cai et al. [18] reported that the protective layer with a certain strength and lubricity, accumulated by CZrP, can quite decrease the friction coefficient and wear rate of CZrP/UHMWPE composites.

Apart from the addition of solid lubricants and hardeners to increase the friction and wear behavior of composites, it is also quite important for researchers to intensively study the surrounding environment (like sliding speed, applied load, testing temperature, etc.) as friction is very complicated [19–21]. Fridrici et al. [22] studied the friction and wear behavior of PTFE coatings under various testing temperatures (25 to 180 °C) and environment (dry and in water/oil); the results showed that the wear rate increase with an increase in temperature and the wear formed under dry, water and oil environment were quite dependent on the tribological behavior of the transfer film on the counterface. Xiong et al. [23] described that the smeared plasma on the worn surface resulted in the lowest wear rate of UHMWPE under plasma compared with dry and water lubrication condition. Tan et al. [24] point out that with the increase in ambient temperature, the wear mechanism of PTFE/Kevlar fabric composites transfer from abrasive to mild and severe adhesive wear/thermal fatigue. Bandaru et al. [25] investigated the effect of hygrothermal aging on the friction and wear resistance of glass/PTFE composites and found that hygrothermal ageing reduced the abrasive wear resistance of materials and the inclusion of steel metal mesh improved the wear resistance. Moreover, Gürgen et al. [26,27] found that oxidation and high temperature lead to a heavy degradation in the anti-wear properties of UHMWPE composites and the addition of Si$_3$N$_4$ and SiC inorganic particle can ease the situation. Wang et al. [28] studied the tribological properties of glass fiber/PTFE materials under different temperatures and vacuum degrees, and described that delamination and fatigue wear are predominant during the sliding process.

As mentioned above, the research work on inorganic fillers-modified polymer composite materials has received extensive attention. The polymer matrix materials, however, still suffer from wear-related problems and there are a few studies that investigate the influence of surrounding temperature. Thus, in this paper, the PTFE composites with different contents of MoS$_2$ were fabricated and the effect of testing temperature from room temperature to 250 °C on the tribological properties of PTFE composites was investigated, and the related wear and lubricating mechanisms were analyzed.

## 2. Experimental Part

### 2.1. Materials Preparation

Firstly, the PTFE powder was dispersed in an absolute ethyl alcohol flask and stirred for 20 min and then, the solid lubricant MoS$_2$ was added and stirred continuously for another 20 min. Secondly, the mixed powder was put into a vacuum drying oven to remove the alcohol; the temperature was set to 60 °C. Then, the mixed materials were put into the stainless steel die and put in a hot press sintering furnace (as shown in Figure 1a); the powders were pressed under 15 MPa for 90 min under 350 °C sintering temperature. Finally, the PTFE matrix composites were obtained. The composition of the composites is listed in Table 1. The MoS$_2$ weight ratio was chosen based on the experimental data which had been carried out in our laboratory. Powders of MoS$_2$ (AR, 99%, Shanghai McLean Biochemical Technology Co., Ltd., Shanghai, China) and PTFE (ML01, Liaocheng Fuer New Material Technology Co., Ltd., Liaocheng, China) were used in this paper.

**Table 1.** Composition of PTFE matrix composites.

| Composites | PTFE (wt.%) | MoS$_2$ (wt.%) |
| --- | --- | --- |
| P | 100 | 0 |
| PM1 | 99 | 1 |
| PM2 | 98 | 2 |
| PM5 | 95 | 5 |

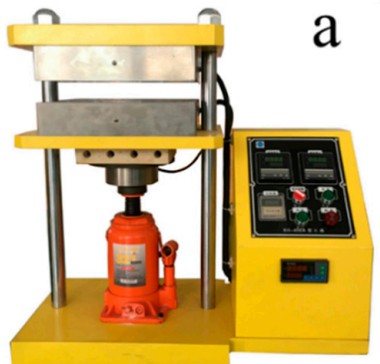 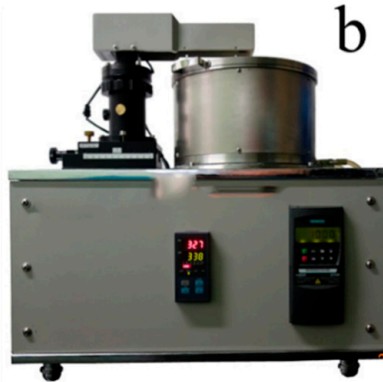

**Figure 1.** Macro photograph of the (**a**) hot pressing sintering equipment (**b**) HT-1000 high temperature tribotester.

### 2.2. Tribological Performance Tests

High-temperature tribometer tester was used to characterize the tribological properties of PTFE matrix composites (HT-1000, as shown in Figure 1b. Lanzhou Zhongke Kaihua Technology Development Co., Ltd., Lanzhou, China). The contact form is pin-on-disk and in a rotation mode, as shown in Figure 2 and the detailed testing parameters are shown in Table 2. The friction coefficient (COF) datum was collected in real time and the ambient temperatures of the samples were measured using a thermocouple which was placed inside the furnace. After the reciprocating friction test, a contact probe (MT-500, Lanzhou Zhongke Kaihua Technology Development Co., Ltd., Lanzhou, China) was used to measure the wear volume, then the wear volume of the composite was measured using a contact probe, and the calculation formula for the volumetric wear rate is as follows:

$$W = \frac{V}{FL}$$

where $W$ is the wear rate (mm$^3$/Nm), $V$ is the wear volume (mm$^3$), $F$ is the load (N) and $L$ is the sliding distance (m). All frictional tests were repeated for at least three times under the same testing condition to ensure the accuracy of data and the average value is taken as the end COF and wear rate.

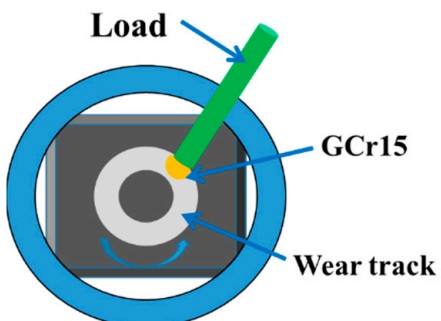

**Figure 2.** Schematic diagram of contact form.

**Table 2.** Tribotest condition of UHMWPE matrix composites.

| Material | P, PM1, PM2, PM5 |
|---|---|
| Testing temperatures | 25 °C, 50 °C, 100 °C, 150 °C, 200 °C, 250 °C |
| Sliding speed | 0.33 m/s |
| Load | 10 N |
| Test time | 30 min |
| Counterpart materials | GCr15 |

*2.3. Characterization*

To observe the morphology features and analyze the composition of worn surface, field emission scanning electron microscopy (FE-SEM, LSM5600LV) and Raman spectroscopy using a LabRAM HR Evolution (Renishaw inVia Reflex, with a 532 nm wavelength) were employed.

## 3. Results and Discussion

*3.1. Friction and Wear Performance Test*

Figure 3 shows the changing trend in COF vs. testing temperature as PTFE matrix composites were coupled with a GCr15 bearing steel ball. The COF decrease continuously with the increase in testing temperatures from 25 to 250 °C. Overall, all four composites present excellent lubricating properties and the variation range is 0.14–0.19. For pure PTFE material, as the testing temperature increases to 100 °C, the COF descends from 0.19 to 0.16, and at 150 °C and above, the value remains around 0.16. For PM2 and PM5 composites, the changing trend in COF is similar to pure PTFE material: from 25 °C to 200 °C, the COF decrease continuously to the lowest value (0.14 for PM2 and 0.15 for PM5), and at 250 °C, the COF increase slightly for PM5. As the SEM images show, the worn surfaces are smooth and there is only some plastic deformation smeared on it, and thus we may deduct that a lubricating film is formed on the worn surface under high surrounding temperatures, so low COF is exhibited. For the PM1 composite, the COF changing trend is slightly different compared with that of PM2/PM5: from room temperature to 50 °C, the COF gradually increases to the highest value of 0.19 and then decreases to the lowest one of 0.15 at 150 °C, and as the testing temperature increases to 200 °C and above, the COF fluctuates around 0.16–0.17. We may explain this phenomenon from the following aspect: as the testing temperature increases from 25 °C to 250 °C, the diffusion rate of solid lubricants from matrix to sliding surface increases and moreover, the mechanical properties of the PTFE matrix decrease, both of which result in the increasing thickness of the lubricating film, thus, the excellent lubricating performance is exhibited [28–30].

As for the effect of $MoS_2$ content, the COF changing trend for the four composites is almost the same. At 25 °C and 150 °C, all four composites show a similar value, while at 50 and 100 °C, the PM1 composite shows the highest value and the datum is basically the same for P/PM2/PM5. As the surrounding temperature increases to 200 and 250 °C, P and PM1 present the highest value, and composites of PM2 and PM5 show the lowest COFs. By comparison, as the testing temperature exceeds 150 °C and the $MoS_2$ contents are 2 wt.% and 5 wt.%, due to the formation of relatively complete tribofilm, these two composites exhibit excellent lubricating performance, and the COF is lower than 0.15 [10].

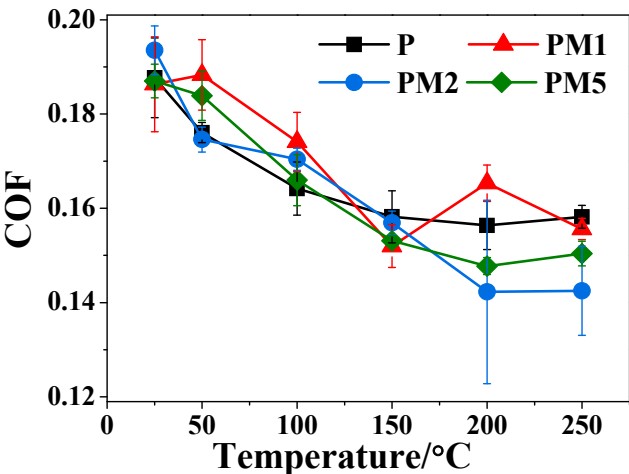

**Figure 3.** Friction coefficients of four samples of PTFE composites as a function of testing temperatures.

Moreover, the typical curve of COF vs. testing time under different testing temperatures are also summarized, as shown in Figure 4. We can clearly see the obvious effect of MoS$_2$ content and testing temperature on the changes of COFs with sliding time as the tribological test processed. At 25 °C (Figure 4a), composites of P, PM1 and PM2 present a short run-in stage (about 0–2.5 min) and during this process, the COF showed an increasing trend with sliding time, then all these three composites maintain a steady COF during the wear stage until the end of the experiment. For the PM5 composite, the COF shows a decreasing trend during the run-in stage (about 0–5 min) and then the value increases slightly with the increase in sliding time till 30 min. At 50 °C (Figure 4b), the COFs vs. sliding times for P, PM2 and PM5 composites become low and steady during the steady stage, which may be attributed to the soften of sliding surface, while the changing trend for PM1 is almost the same compared with that at 25 °C. As the testing temperature increases to 100 °C (Figure 4c), all four composites present a decreasing trend during the run-in stage (about 0–2.5 min). In the steady stage, the COF vs. sliding time decrease continuously for P and PM1 and the trend is steady for PM2 and PM5. At 150 °C (Figure 4d), the COFs of the composites are relatively high at the beginning of the sliding process, while as the sliding time increases, the values decrease to the lowest and is maintained till 30 min. Additionally, the run-in stage is about 10 min for the P, PM1 and PM5 composites, while for PM2, the changing trend is similar to that at 100 °C. As the testing temperature increases to 200 and 250 °C (Figure 4e,f), high COF presents at the beginning of the run-in stage, especially for the P, PM2 and PM5 composites; this may be attributed to the high viscosity of the materials at high temperatures. As the sliding process enters the steady stage, PM2 and PM5 show a continuously decreasing trend in the COF value at 250 °C, while the changing trend for the other composites is similar to that at 100 °C.

In sum, we can say that the fluctuations in COF vs. sliding time for all four composites are low and stable (Figure 4), and we may explain this phenomenon from the following aspect: during the run-in stage, the viscosity between CGr15 and PTFE composites increase with increasing testing temperature, so a relatively high COF is exhibited at the beginning of the run-in stage, while as the sliding process continues, the continuous lubricating film is formed between the contact surfaces, which can be also reflected from the morphologies of the worn surface, and thus, low friction is present during the steady stage.

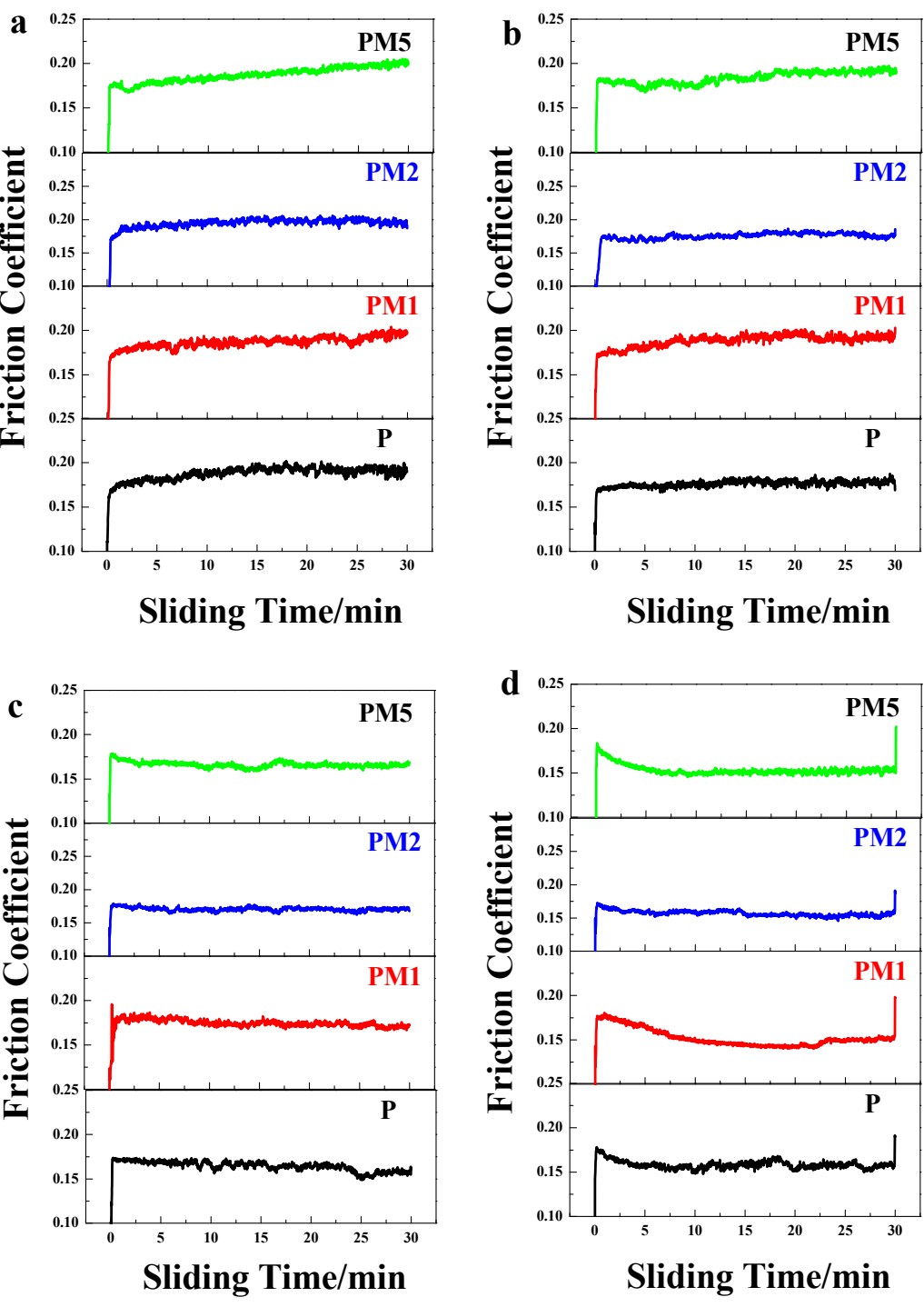

**Figure 4.** *Cont.*

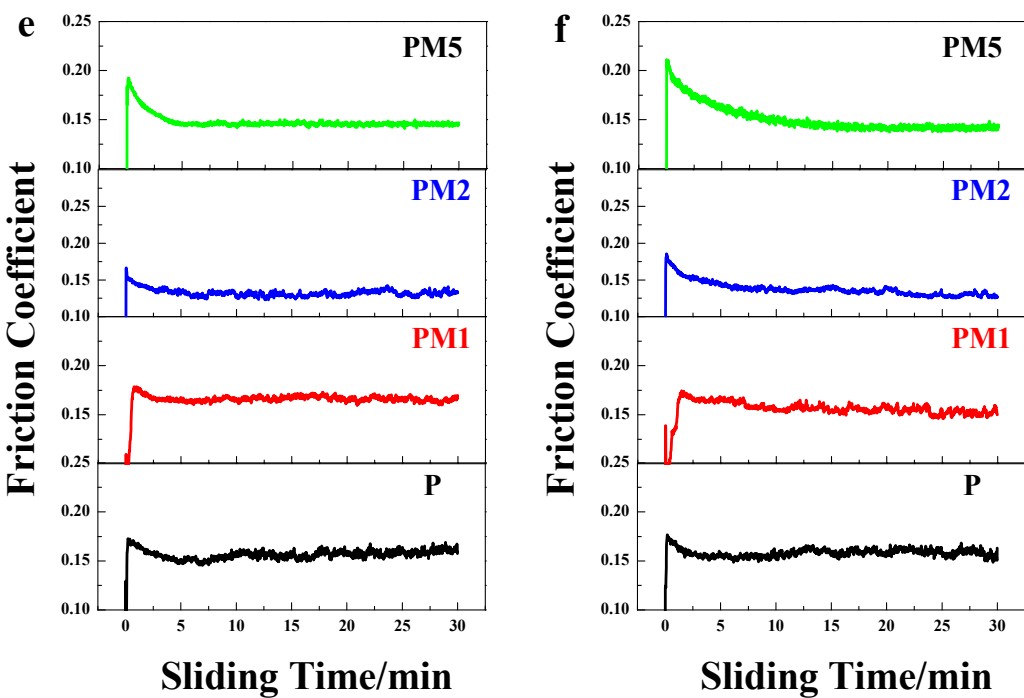

**Figure 4.** The typical friction curve of four composites at different testing temperatures: (**a**) 25 °C, (**b**) 50 °C, (**c**) 100 °C, (**d**) 150 °C, (**e**) 200 °C, (**f**) 250 °C.

The wear rate variation in the PTFE matrix composites with testing temperature is plotted in Figure 5. It can be seen that the wear rate shows a gradually increasing trend from 25 °C to 150 °C for the P, PM2 and PM5 composites, then the value decreases to the lowest one for the PM2 and PM5 composites (PM2: $7.87 \times 10^{-4}$ mm$^3$/Nm and PM5: $4.18 \times 10^{-4}$ mm$^3$/Nm) as the testing temperature increases to the highest value of 250 °C, while for pure PTFE materials, the changing trend is the opposite and it presents the worst wear resistance properties at this temperature ($11.49 \times 10^{-4}$ mm$^3$/Nm). We may deduct that at high testing temperatures, the worn surface softens and the diffusion rate of MoS$_2$ from matrix to sliding surface increases, so a relative complete tribofilm is formed on the contact surface and thus results in the lowest wear rate. For the PM1 composite, the wear rate vs. testing temperature changing trend is different: from 25 to 100 °C, the value shows an increasing trend, then it gradually decreases to the minimum value at 150 °C ($7.36 \times 10^{-4}$ mm$^3$/Nm), while as the testing temperature increases to 250 °C, the datum quite increases to the largest one of $11.49 \times 10^{-4}$ mm$^3$/Nm. This may be attributed to the decreased hardness and strength of the composites as friction heat on the worn surface increases at high testing temperatures, which accelerates fatigue shedding, and ultimately leads to an increase in wear rate [31].

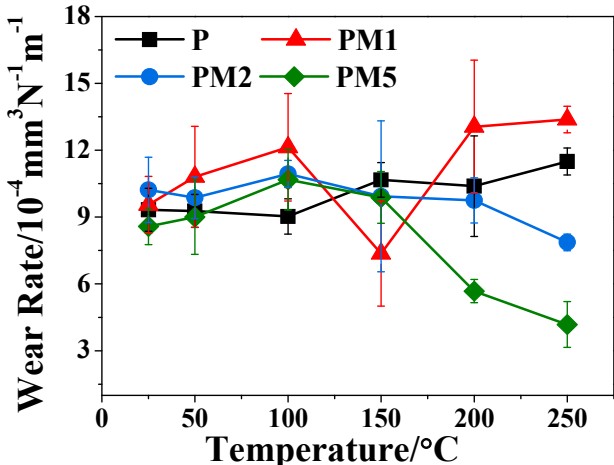

**Figure 5.** Wear rate of PTFE matrix composites as a function of testing temperatures.

Moreover, all four composites show similar wear rates at 25 and 50 °C and the influence of $MoS_2$ content on the values is negligible. As the testing temperature increase to 100 and 150 °C, the wear rate for PM2 and PM5 composites are almost the same, while PM1 presents the highest one at 100 °C ($12.13 \times 10^{-4}$ mm³/Nm) and lowest ($7.36 \times 10^{-4}$ mm³/Nm) at 150 °C; for pure PTFE materials, however, it is the opposite (100 °C: $9.02 \times 10^{-4}$ mm³/Nm; 150 °C: $10.67 \times 10^{-4}$ mm³/Nm). At 200 °C and above, high content of $MoS_2$ results in the increase in lubricating layer under dry sliding condition, so PM5 presents the lowest value (200 °C: $5.68 \times 10^{-4}$ mm³/Nm; 250 °C: $4.18 \times 10^{-4}$ mm³/Nm), followed by PM2 (200 °C: $9.75 \times 10^{-4}$ mm³/Nm; 250 °C: $7.87 \times 10^{-4}$ mm³/Nm), P (200 °C: $10.39 \times 10^{-4}$ mm³/Nm; 250 °C: $11.49 \times 10^{-4}$ mm³/Nm) and PM1 (200 °C: $13.06 \times 10^{-4}$ mm³/Nm; 250 °C: $13.38 \times 10^{-4}$ mm³/Nm); this can be also reflected from the cross-section profiles of the wear rate at different testing temperatures, as shown in Figure 6a–d. Combined with COF, we may deduce that the optimum $MoS_2$ content is 2 wt.%.

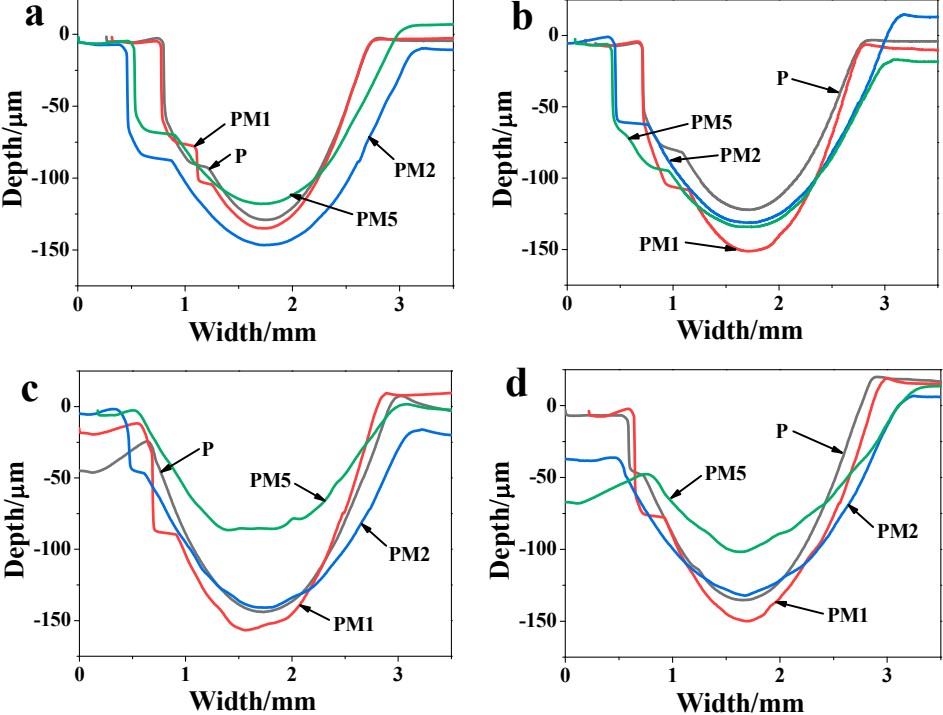

**Figure 6.** Cross-section profiles of the wear rate for four composites at different testing temperatures: (**a**) 25 °C, (**b**) 50 °C, (**c**) 200 °C, (**d**) 250 °C.

### 3.2. Worn Surface Analysis

In order to further investigate the friction and wear behavior of the PTFE matrix composites, SEM observation was conducted to analyze the morphologies of the worn surface, as shown in Figures 7–15. Figures 7–9 present the macro-SEM photos of the P, PM1 and PM2 composites, respectively, at different testing temperatures. Clearly, all three composites showed smooth worn surfaces at 25 (Figures 7a, 8a and 9a) and 50 °C (Figures 7b, 8b and 9b) in the whole, while as the sliding temperature increases to high temperatures (200 and 250 °C), the characteristic was different: for pure PTFE material, there is almost no worn surfaces and the morphologies is almost the same compared with that at 25 and 50 °C, while for the PM1 and PM2 composites, flaking pits and smeared wear debris appeared on the worn surface, indicating that fatigue and abrasive wear are the main wear mechanisms.

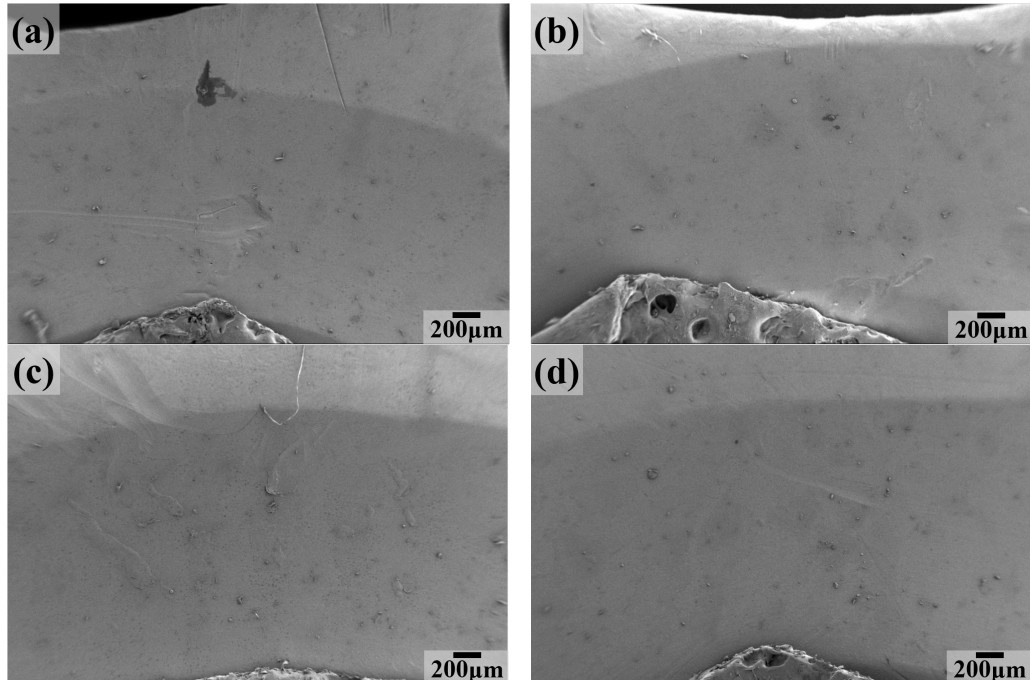

**Figure 7.** SEM images of the worn surface for the P composite at different temperatures: (**a**) 25 °C, (**b**) 50 °C, (**c**) 200 °C, (**d**) 250 °C.

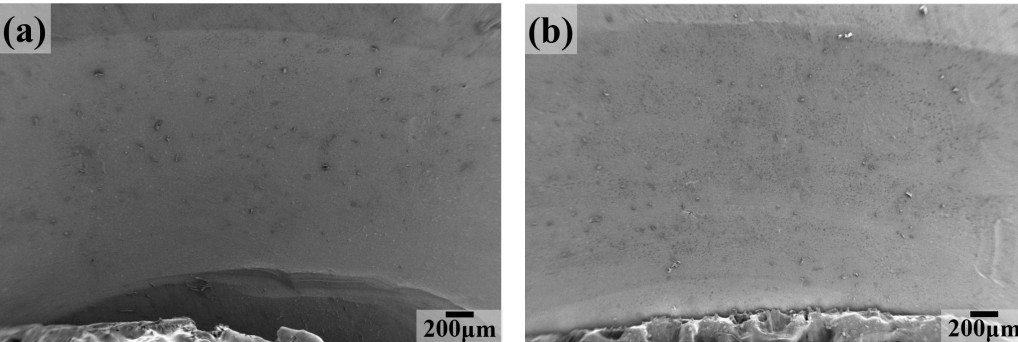

**Figure 8.** *Cont.*

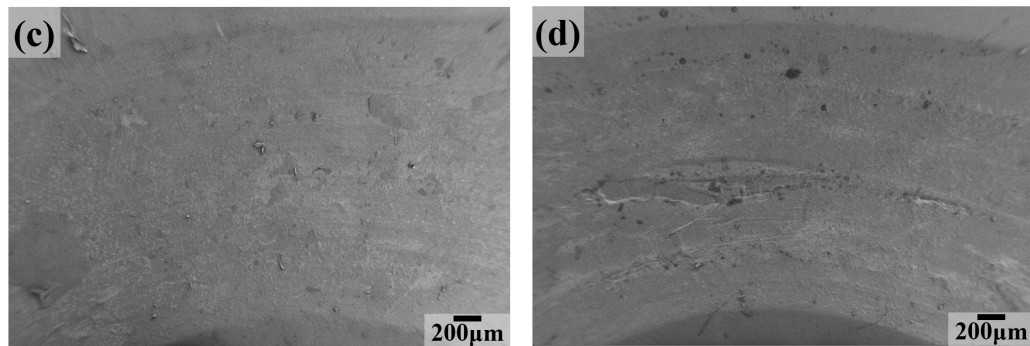

**Figure 8.** SEM images of the worn surface for the PM1 composite at different temperatures: (**a**) 25 °C, (**b**) 50 °C, (**c**) 200 °C, (**d**) 250 °C.

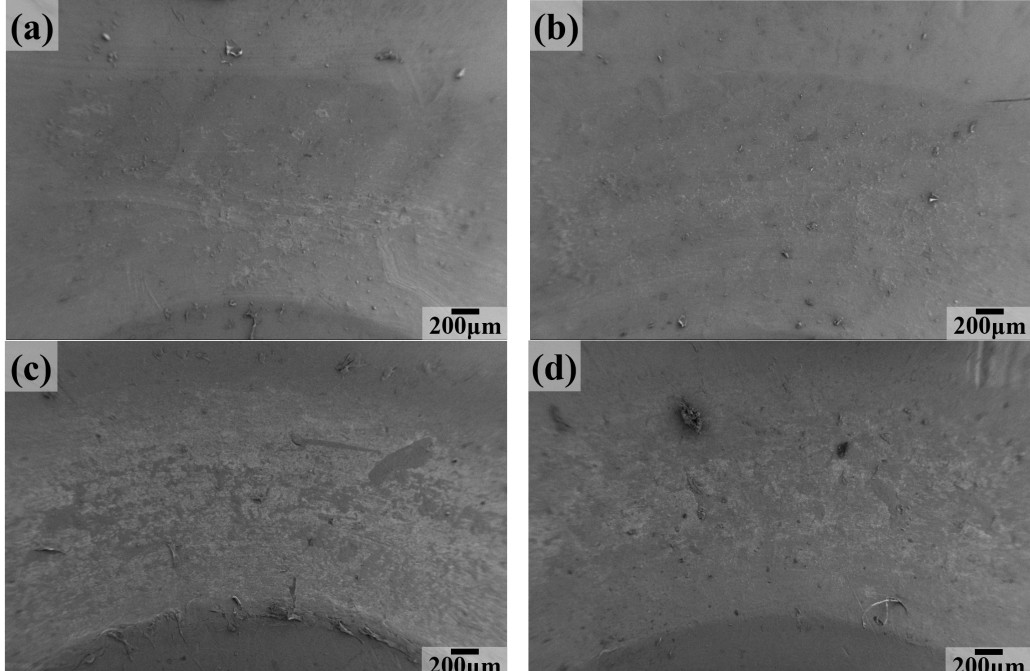

**Figure 9.** SEM images of the worn surface for the PM2 composite at different temperatures: (**a**) 25 °C, (**b**) 50 °C, (**c**) 200 °C, (**d**) 250 °C.

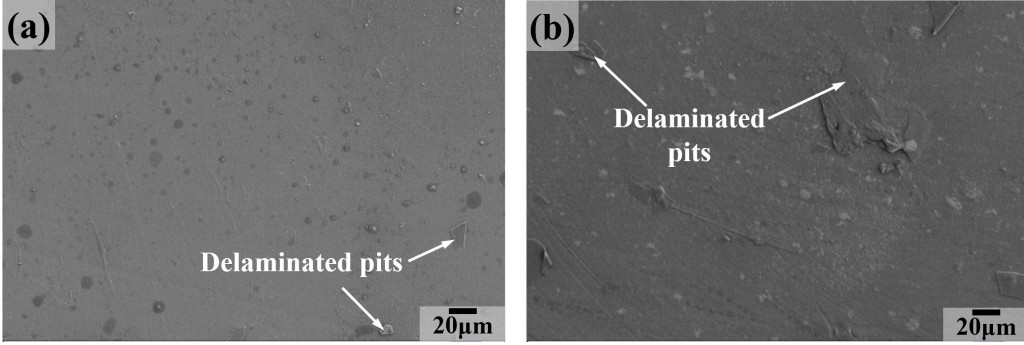

**Figure 10.** *Cont*.

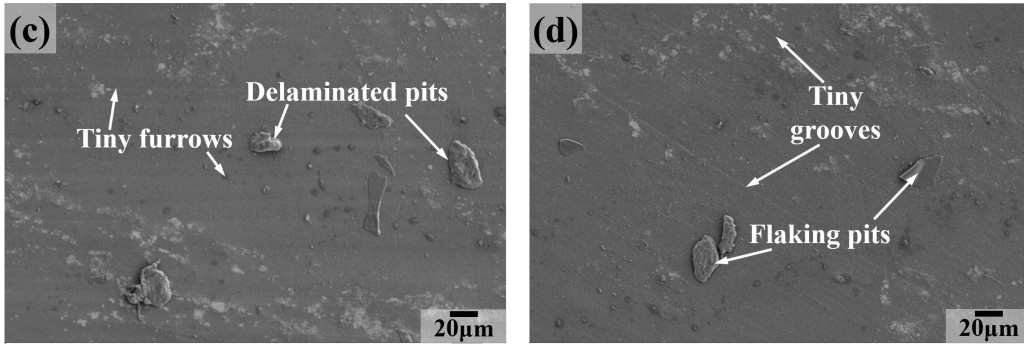

**Figure 10.** SEM images of the worn surface at 25 °C: (**a**) P, (**b**) PM1, (**c**) PM2, (**d**) PM5.

Figures 10–15 present the micro-SEM images of the P, PM1 and PM2 composites at different testing temperatures from 25 to 250 °C. Figure 10 illustrates the worn surface morphologies of the four composites at 25 °C. For pure PTFE materials, the worn surface is characterized by a smooth layer and there are few flaking pits smeared on it (Figure 10a), which is a sign of slight abrasive wear. As the $MoS_2$ content increased to 1 wt.%, 2 wt.% and 5 wt.%, tiny furrows appeared on the worn surface, except some flaking pits (Figure 10b–d), which indicates that the main wear mechanisms are fatigue and abrasive wear.

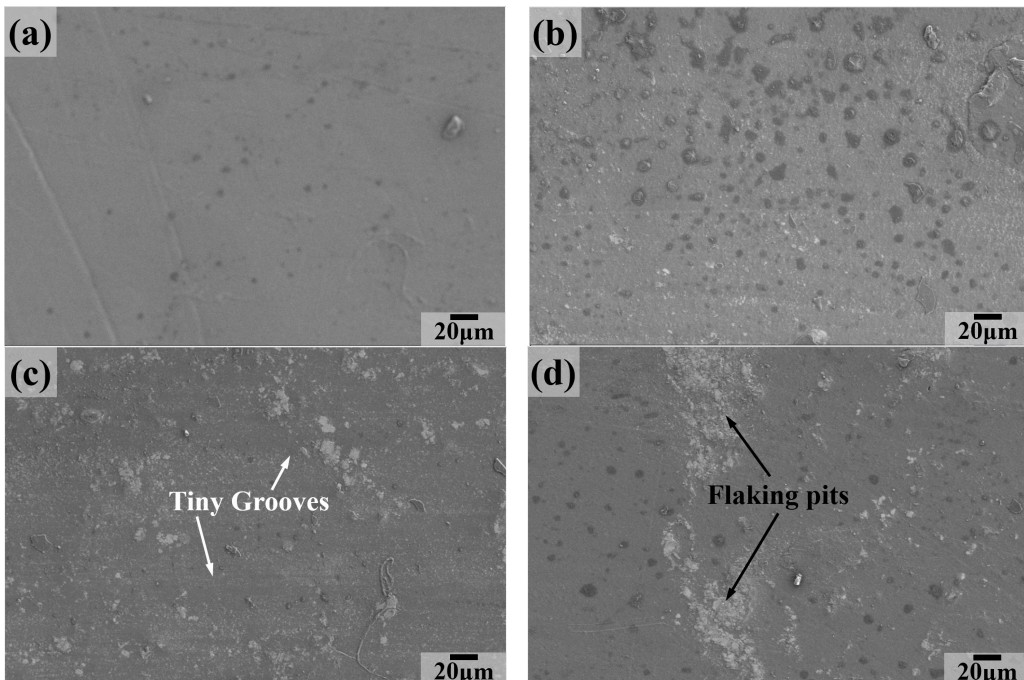

**Figure 11.** SEM images of the worn surface at 50 °C: (**a**) P, (**b**) PM1, (**c**) PM2, (**d**) PM5.

SEM images of the worn surface for all the four composites at 50 °C is shown in Figure 11. The worn surface morphology is almost the same for pure PTFE materials compared with that at 25 °C, both characterized by a smooth layer. For the PM1, PM2 and PM5 composites, tiny furrow with some little flaking pits instead of large deformation appeared on the worn surface, which suggests that slight abrasive wear is the main wear mechanism; this is consistent with the variation in COF and wear rate.

Figures 12–15 illustrate the SEM images of the worn surface at testing temperatures of 100 °C, 150 °C, 200 °C and 250 °C. For pure PTFE materials (Figures 12a, 13a, 14a and 15a), smooth lubricating layers with some delaminated pits and grooves appear, indicating that the main wear mechanism is abrasive wear. Moreover, the grooves become deep and obvious with the increase in testing temperature, which is consistent with the changing trend in wear rate (Figure 5). For the PM1 composite, large grooves appeared on the worn surface at 100 °C (Figure 12b), while as the testing temperature increases to 150 °C and above (Figures 13b, 14b and 15b), smeared flaking pits and grooves appear to be the main features, suggesting that abrasive and fatigue wear are the main wear mechanisms from 100 to 250 °C. As described above, the COFs and wear rate decrease with the increase in testing temperatures for the PM2 and PM5 composites. As shown in Figure 12c,d, we can clearly see that the small delaminated pits and grooves are the main features at 100 °C, while as the testing temperature increases to 150, 200 °C and 250 °C, large white and grey worn scar appeared on the worn surface (Figure 13c,d, Figure 14c,d and Figure 15c,d); we deduced that this can be attributed to the smeared large flaking pits which peel off from the matrix composites, so the main wear mechanisms are abrasive and fatigue wear.

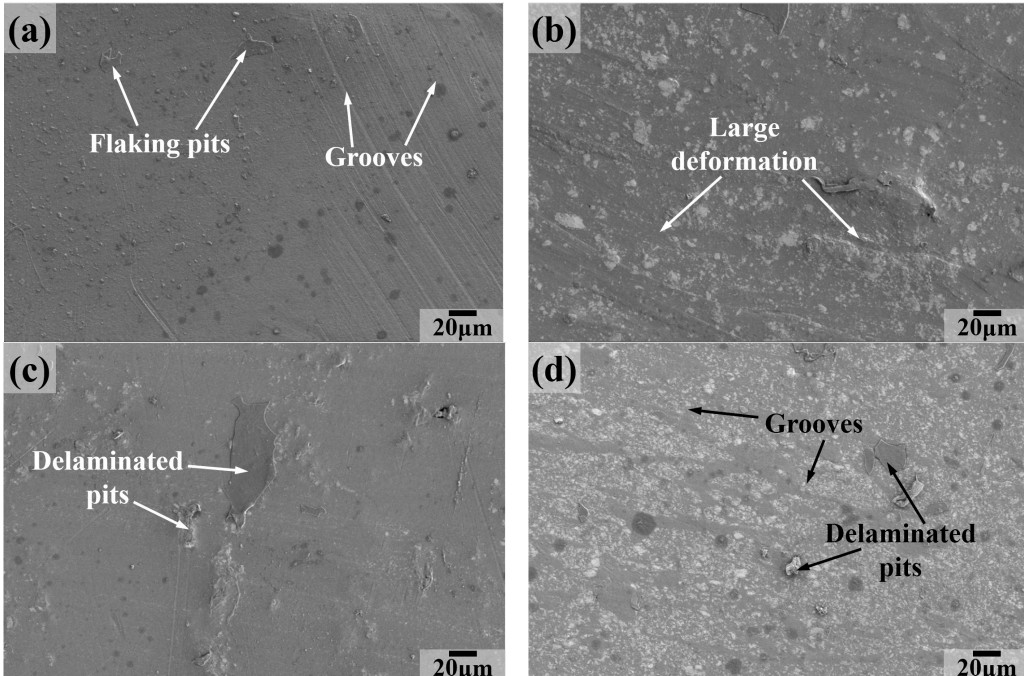

**Figure 12.** SEM images of the worn surface at 100 °C: (**a**) P, (**b**) PM1, (**c**) PM2, (**d**) PM5.

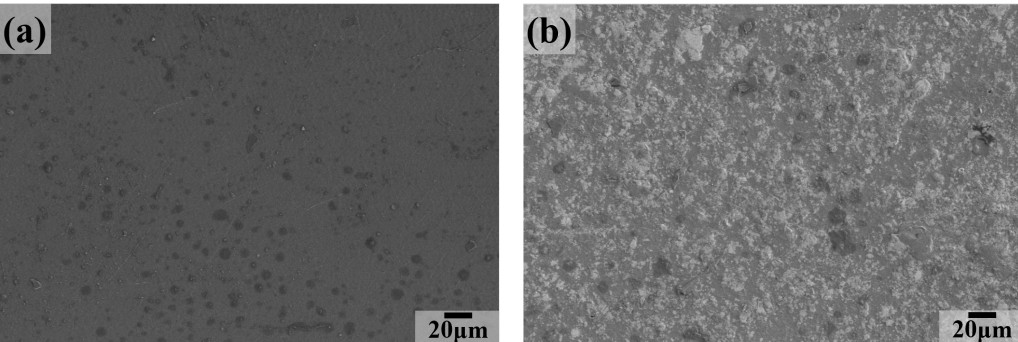

**Figure 13.** *Cont*.

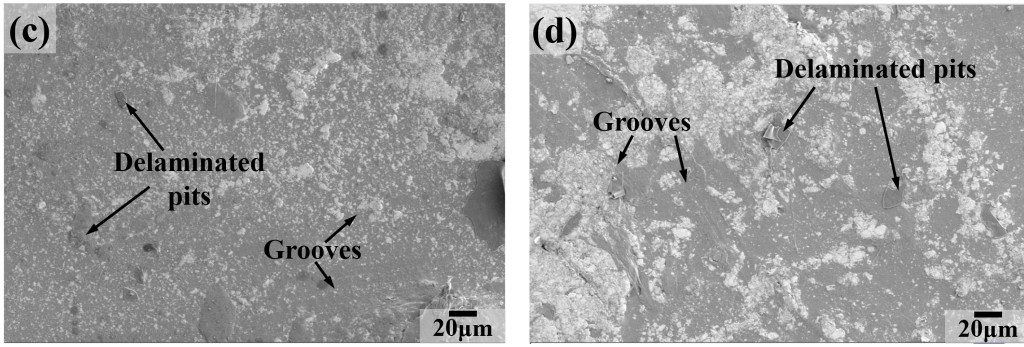

**Figure 13.** SEM images of the worn surface at 150 °C: (**a**) P, (**b**) PM1, (**c**) PM2, (**d**) PM5.

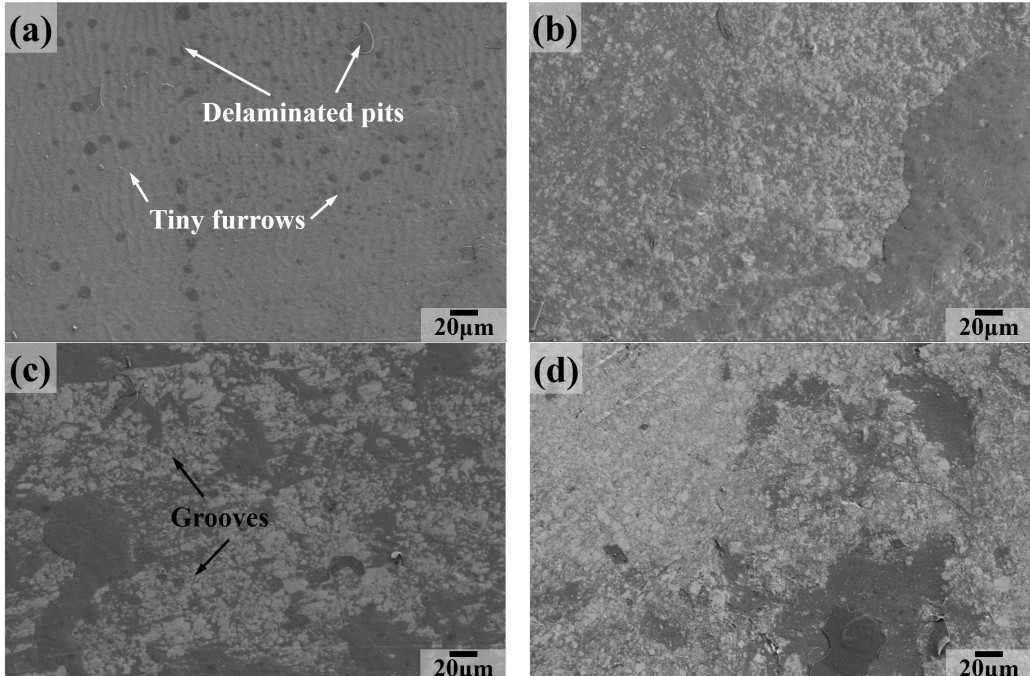

**Figure 14.** SEM images of the worn surface at 200 °C: (**a**) P, (**b**) PM1, (**c**) PM2, (**d**) PM5.

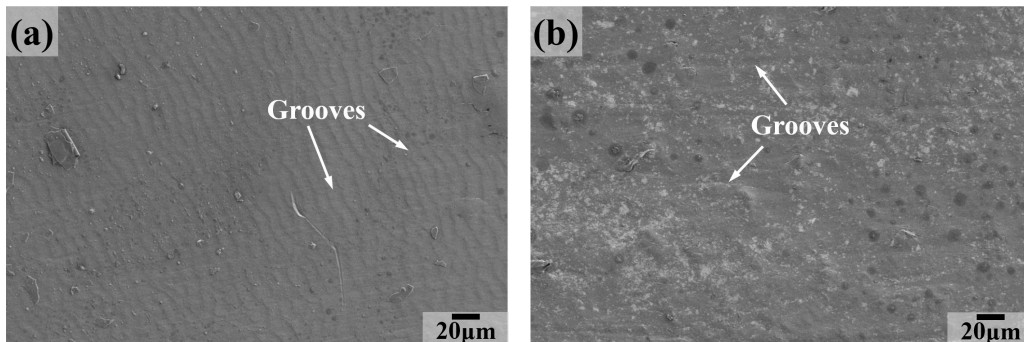

**Figure 15.** *Cont*.

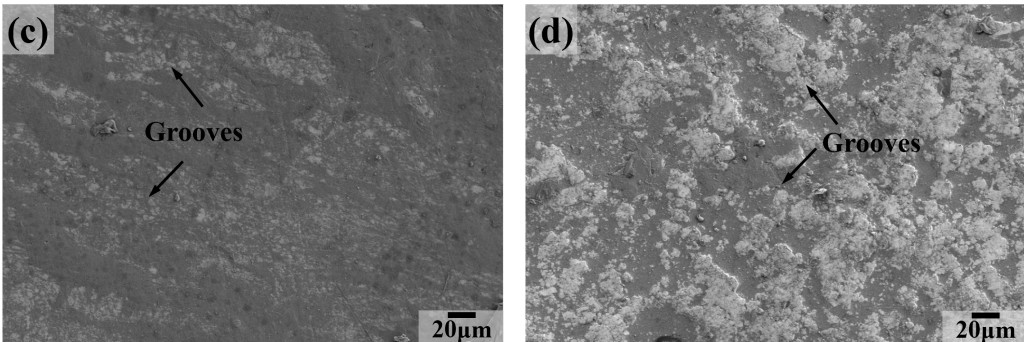

**Figure 15.** SEM images of the worn surface at 250 °C: (**a**) P, (**b**) PM1, (**c**) PM2, (**d**) PM5.

As we all know that the friction and wear performances of composites are quite dependent on the surface composition during the sliding process. In order to further analyze the influence of testing temperature and $MoS_2$ content on the friction and wear properties of the PTFE composite materials, Raman spectra of the sliding surfaces for the composites were analyzed, as shown in Figures 16 and 17. Figure 16 presents the Raman spectrum of the worn surface after the sliding process for the four composites at 250 °C. We can say that the relative intensity of $MoS_2$ on the worn surface increases with the increasing content of solid lubricant. Combined with COFs and wear rate variation (Figures 3 and 5), we may deduct that for the PM1 composite, only some $MoS_2$ solid lubricant is smeared on the worn surface, while as the $MoS_2$ content increases to 2 wt.% and 5 wt.%, more solid lubricant diffused from the matrix to the worn surface during the sliding process, so it exhibited a relatively low COF and wear rate [32].

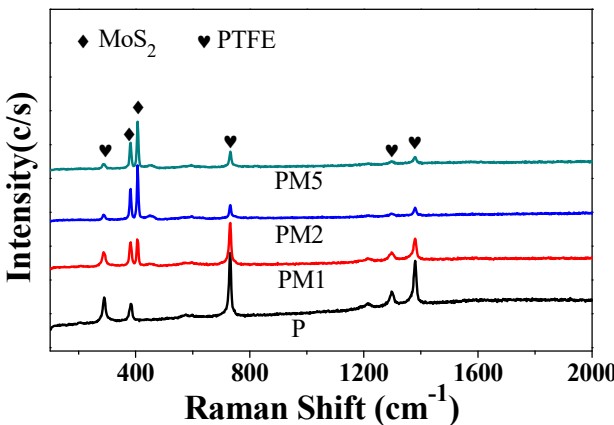

**Figure 16.** Raman spectrum of the worn surfaces at 250 °C for the four composites.

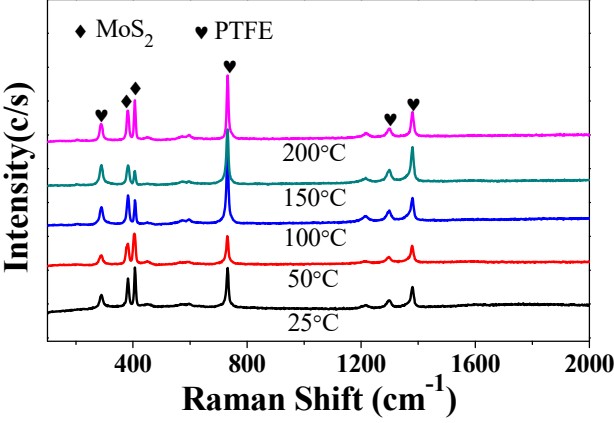

**Figure 17.** Raman spectrum of the worn surfaces at different testing temperatures for PM2.

Figure 17 reflects the Raman spectrum of the worn surface for the PM2 composite at different testing temperatures from 25 °C to 200 °C, respectively. Clearly, the relative intensity of the $MoS_2$ peak on the worn surface increases, except at 150 °C, with the increase in testing temperature. Combined with the changing trend in COF and wear rate, we can deduce that high temperatures can accelerate the diffusion rate of $MoS_2$ solid lubricant and the relative smooth worn surface can also prove this phenomenon (Figures 13–15), therefore, the PM2 composite exhibits relatively lower wear rate and COF than the other three composites [33].

In a comprehensive comparison, the testing temperature and $MoS_2$ content have a significant effect on the tribological behavior of the PTFE matrix composite. As reported, the wear performance of the composites is quite dependent on the consumption and formation rate of the lubricating film on the sliding surface. Generally, as the diffusion rate of solid lubrication is greater than the consumption rate, more lubricating films will form on the worn surface, thus exhibiting a low friction and wear rate, while when the conditions are reversed, high COF and wear rate may be presented. We can explain this wear performance of the composites at different testing temperatures from the following perspective: as the testing temperature increases from 25 °C to 250 °C, more and more solid lubricant diffuses from the matrix to the worn surface and the thickness of the lubricating film covered on the materials increases. Moreover, during subsequent contact, only a part of the lubricating film is removed, especially as with temperatures up to 150 °C and above, so the COF decreases with the increase in testing temperatures, especially for the composites of PM2 and PM5 [34]. For the fluctuation trend in the wear rate, as the testing temperature increases from 25 °C to 250 °C, on one hand, the increasing thickness of lubricating film will protect the worn surface from being destroyed during the shearing process, and on the other hand, the deteriorative mechanical properties will decrease the wear resistance of the composites. Combining these two mentioned conditions, high content of solid lubricants results in the increasing thickness of the lubricating film, so a relatively low wear rate is exhibited for the PM2 and PM5 composites at 200 and 250 °C [29].

## 4. Conclusions

The PTFE matrix composites with different contents of $MoS_2$ (1 wt.%, 2 wt.% and 5 wt.%) were prepared using the hot-pressed sintering method, and their frictional behavior under different testing temperatures (25–250 °C) were investigated. The conclusions drawn from this work are as follows.

(1) Generally, all four composites show excellent tribological performance under the testing temperatures from 25 to 250 °C. The COF and wear rate are approximately (0.14–0.19) and (4.18–13.38 $\times$ $10^{-4}$ $mm^3$/Nm);

(2) Adding $MoS_2$ solid lubricating into the PTFE matrix could increase the lubricating and the wear resistance properties of PTFE composites, especially at high temperatures (200 and 250 °C). Generally, the COF in the mass decreases with the increase in testing temperatures, while the wear rate remains at a stable value. As for the effect of the $MoS_2$ content, from 25 °C to 150 °C, all four composites show a similar COF and wear rate, while at 200 and 250 °C, P and PM1 present the highest value, and composites of PM2 and PM5 show the best wear resistance and lubricating properties. By comparison, the optimal content is 2 wt.%.

(3) Both solid lubricating and testing temperature play essential roles in the variations of wear mechanism. At low testing temperatures (25–100 °C), the worn surfaces are relatively smooth and the main wear mechanism is slight abrasive wear. From 150 °C to 250 °C, large delamination and grooves appeared on the worn surface and the main wear mechanism transformed to fatigue and severe abrasive wear.

**Author Contributions:** Conceptualization, J.Z. and Y.H.; methodology, J.Z.; software, H.W.; validation, Z.J. (Zhenguo Jiang), R.Z. and. L.W.; formal analysis, J.Z.; investigation, Y.H. (Yuqiang Huang); resources, Z.J. (Zhengfeng Jia); data curation, J.Z.; writing—original draft preparation, J.Z.; writing—review and editing, J.Z.; visualization, Y.H. (Yunxiang Han); supervision, H.W.; project administration, J.Z.; funding acquisition R.Z. All authors have read and agreed to the published version of the manuscript.

**Funding:** This work was supported by the National Natural Science Foundation of China (52105190); Shandong Province Science and Technology Small and Medium Enterprises Innovation Ability Improvement Project (2022TSGC1220, 2022TSGC2575) and Liaocheng Small and Medium-sized Enterprise Climbing Plan (2022PDJH25, 2022PDJH22).

**Data Availability Statement:** Not applicable.

**Conflicts of Interest:** The authors declare no conflict of interest.

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
