# Peer review of "High Temperature Friction and Wear Behavior of PTFE/MoS2 Composites"

_lubricants, doi:10.3390/lubricants11080312_

Round 1

Reviewer 1 Report

The manuscript reports on composites formed by polytetraethylene and MoS2, as potentially promising materials for lubrication at high temperatures.

The manuscript can be reconsidered for acceptance, after proper revision. Below are comments and suggestions that should help the authors improve it.

1\ Author list: please check how Author Four is presented (there seems to be an extra comma in the name)

2\ Affiliations 2,3 need addresses, please add

3\ On page 1, there is discrepancy between author names in line 4 and author names in the left part /margin of the page (double check the order of names).

4\ Figure 1 is not clear to reader.

 5\ Figure 3: the figure needs labels (a), (b), (c), (c), (e), (f). Plus their description must be added to the caption

6\ Figure 4: it is not clear why sample PM1 behaves differently. Namely, while the other samples do not demonstrate any anomalous behavior at 200 degrees, only sample PM1 shows a drop in wear rate at this temperature. why?

7\ Probably, the most important comment: The novelty and originality of this manuscript are not clearly stated in the text. The authors must clearly mention: what was previously published/ investigated (by others) in the PFTE-MoS2 system. What did the other researchers publish before? Is this the first report on this system?

8\ Conclusions: It is not clear in this section, what was really new and original in the present manuscript/work in comparison with previous works on similar composite materials.

English must be improved, there are too many cased of poor language:

1\ line 28: must be improved

2\ lines 41-42 , 44-45: language must be improved

3\ line 39: fibre ; compare with line 55: fiber

4\ lines 51-52 and lines 60-61: improve language

5\ line 52: were to mild ("wear to mild" ?)

6\ line 68: improve English

7\ Page 3, Table 2: replace commas in the table, some commas seem to belong to Asian fonts (in the row: P, PM1, PM2, PM5)

8\ lines 117, 122, 125: language must be improved

9\ line 121: influence MoS2  (influence of MoS2? )

10\ lines 235 and 244: Why the word "spectrum" is spelled with capital letter?

11\ lines 146 and 150: language must be improved

Author Response

For reviewer 1

Dear reviewer,

Thank you very much for your perfect comments on our manuscript. We revised our manuscript according to your suggestions. Your comments were replied as follow.

General comments: The manuscript can be reconsidered for acceptance, after proper revision. Below are comments and suggestions that should help the authors improve it.

Response: Thank you very much for giving us a chance to revise our manuscript and consider the revised version.

Comment 1: Author list: please check how Author Four is presented (there seems to be an extra comma in the name).

Response: Thank you for your suggestions, the author’s name had been corrected and the variation is marked in the revised manuscript.

Comment 2: Affiliations 2, 3 need addresses, please add.

Response: Thank you for your suggestion, the addresses of Affiliations 2,3 were added and the variation is marked in the revised manuscript.

Comment 3: On page 1, there is discrepancy between author names in line 4 and author names in the left part /margin of the page (double check the order of names).

Response: Thank you for your suggestions, the author names had been checked carefully.

Comment 4: Figure 1 is not clear to reader.

Response: Thank you for your suggestion, Fig. 1 showed the schematic diagram of contact form and the variation is marked in the revised manuscript.

Comment 5: Figure 3: the figure needs labels (a), (b), (c), (c), (e), (f). Plus their description must be added to the caption.

Response: Thank you very much for your suggestions, the labels of (a), (b), (c), (c), (e), (f). plus their description were added and the variation is marked in the revised manuscript.

Comment 6: Figure 4: it is not clear why sample PM1 behaves differently. Namely, while the other samples do not demonstrate any anomalous behavior at 200 degrees, only sample PM1 shows a drop in wear rate at this temperature. why?

Response: Thank you very much for your questions. This may be attributed to the decreased hardness and strength of the composites as friction heat on the worn surface increase at high testing temperature, especially for PM1 composites, which accelerates fatigue shedding, and ultimately leads to an increase in wear rate. The discussion about these results was enriched and the changes had been marked in the revised manuscript.

Comment 7: Probably, the most important comment: The novelty and originality of this manuscript are not clearly stated in the text. The authors must clearly mention: what was previously published/ investigated (by others) in the PFTE-MoS2 system. What did the other researchers publish before? Is this the first report on this system?

Response: Thank you for your suggestions. The previously published/ investigated (by others) in the PFTE-MoS2 system were searched carefully and the results were added in the instruction and the variation is marked in the revised manuscript.

Comment 8: Conclusions: It is not clear in this section, what was really new and original in the present manuscript/work in comparison with previous works on similar composite materials.

Response: Thank you very much for your questions. The really new and original in the present manuscript/work in comparison with previous works on similar composite materials is the investigation of high temperature friction and wear behavior of PTFE-MoS2 composites.

Comments on the Quality of English Language

English must be improved, there are too many cased of poor language:

Comment 9: line 28: must be improved

Comment 10: lines 41-42, 44-45: language must be improved

Comment 11: line 39: fibre; compare with line 55: fiber

Comment 12: lines 51-52 and lines 60-61: improve language

Comment 13: line 52: were to mild ("wear to mild" ?)

Comment 14: line 68: improve English

Comment 15: Page 3, Table 2: replace commas in the table, some commas seem to belong to Asian fonts (in the row: P, PM1, PM2, PM5)

Comment 16: lines 117, 122, 125: language must be improved

Comment 17: line 121: influence MoS2 (influence of MoS2? )

Comment 18: lines 235 and 244: Why the word "spectrum" is spelled with capital letter?

Comment 19: lines 146 and 150: language must be improved

Response to 9-19: Thank you very much for your valuable suggestions, the English has been carefully checked and revised throughout paper.

Reviewer 2 Report

The authors presented an article about “High temperature friction and wear behavior of PTFE/MoS2 composites.”  

In this study, the authors carried out wear tests at different temperatures for the composite materials they produced by adding MoS2 at different weight ratios to the PTFE main matrix. Although the subject of the study is current and remarkable, there are some shortcomings. I think the authors did not express very well how these composites are used today in which industrial area and on which problem they are doing this study. But in general, I can say that I find the work valuable and successful.

I think the paper is not well organized and appropriate for the “Lubricants” journal, but the paper will be ready for publication after major revision.

·       The abstract don’t looks good. A brief description of the problem of the study and the brief importance of solving the problem should be included. The paper's originality needs to be clarified further in the abstract. Please, involve the novelty of this paper, not what you have done in this study.Also,  please include all significant numerical results.

·       What is the problem? Why was the manuscript written? Please explain the reason in the introduction part. In the last paragraph of the introduction, the novelty of the study and the differences from the past in detail should be expressed.

·       Literature review was insufficient. Please support with current studies.

·       What was used as inert gas during hot pressing?

·       How did the authors choose the MoS2 weight ratio?

·       On what basis was the wear test data selected? Please explain.

·       Which international standards were tested and analyzed according to? Please specify.

·       Please add a figure containing the macro photograph of the test instruments and the produced materials to the material method section.

·       The application of ambient temperatures in the experiment is not clearly stated. How did the authors obtain and measure temperatures?

·       Write the wear rate in the material and method section, not as an expression but as an equation.

·       The characterization part was very short. Expand with details.

·       "We may explain this phenomenon from the following aspect: as the testing temperature in-crease from 25 °C to 250 °C, the diffusion rate of solid lubricants from matrix to sliding surface increase and moreover, the mechanical properties of PTFE matrix decrease , both of the above two situation result in the increase thickness of lubricating film, thus the COFs decrease to the minimum value [22-24]." In his statement, how did the authors come to such a conclusion when they did not examine the mechanical properties? Please explain.

·       The graph of COF values at different temperatures is not comparable. It can be more understandable if different graphics are created for each composite. For example, a graph for PM1 composite should include COF values at different temperatures.

·       Explain the reason for the negligibility of MoS2 content at low temperatures in the paper.

·       Since the experiments were repeated three times, please add the error rate to the graphs.

·       Discussions about the results of Raman analysis remained incomplete. For example, explain which molecular structure the peaks occur at which wavelength. Also, support the peaks with literature information.

·       What is the reason for the analysis in Figure 13? Please explain the results in detail.

·       Wear and friction do not happen on one material. The counter bodies must be analyzed (wear appearances). It would be interesting to know whether the counter body sacrifices for the body's better behavior or whether some materials were transferred from the body to the counter body or vice-versa.

·       SEM and EDS figures did not discuss anti-wear and anti-friction mechanisms well. Please, explain the responsible mechanisms clearly. This is one of the critical discussions this manuscript has to have in the results and discussion to show the study's novelty.

·       The conclusions are very weak. The findings should be restored with the correct evidence of the results obtained.

·       Please fix the typographical and eventual language problems in the paper.

*** Authors must consider them properly before submitting the revised manuscript. A point-by-point reply is required when the revised files are submitted.

Author Response

Dear reviewer,

Thank you very much for your carefully checks on our manuscript and giving us very valuable comments and suggestions. Responses to your comments in detail are listed below.

General comments: The authors presented an article about “High temperature friction and wear behavior of PTFE/MoS2 composites.” In this study, the authors carried out wear tests at different temperatures for the composite materials they produced by adding MoS2 at different weight ratios to the PTFE main matrix. Although the subject of the study is current and remarkable, there are some shortcomings. I think the authors did not express very well how these composites are used today in which industrial area and on which problem they are doing this study. But in general, I can say that I find the work valuable and successful. I think the paper is not well organized and appropriate for the “Lubricants” journal, but the paper will be ready for publication after major revision:

Response: Thank you very much for giving us a chance to revise our manuscript and consider the revised version. Due to the low friction coefficient and excellent chemical resistance, the PTFE composite is used in sliding bearings, piston, rings, and gaskets et al. However, PTFE has a poor wear resistance, leading to early failure and leakage problems in seals.

Comment 1: The abstract don’t looks good. A brief description of the problem of the study and the brief importance of solving the problem should be included. The paper's originality needs to be clarified further in the abstract. Please, involve the novelty of this paper, not what you have done in this study. Also, please include all significant numerical results.

Response: Thank you very much for your suggestions, the abstract was enriched and the changes had been marked in the revised manuscript.

Comment 2: What is the problem? Why was the manuscript written? Please explain the reason in the introduction part. In the last paragraph of the introduction, the novelty of the study and the differences from the past in detail should be expressed.

Response: Thank you for your suggestions. Here, the intended applications of the materials are the long-term component, such as snowboard, high performance sails, etc.

Comment 3: Literature review was insufficient. Please support with current studies.

Response: Thank you very much for your valuable suggestion. The literature review was enriched and the changes had been marked in the revised manuscript.

Comment 4: What was used as inert gas during hot pressing?

Response: Thank you very much for your question. There is no inert gas was used during hot pressing process.

Comment 5: How did the authors choose the MoS2 weight ratio?

Response: Thank you for your valuable questions, the MoS2 weight ratio was choose based on the experimental data which had been done in our laboratory.

Comment 6: On what basis was the wear test data selected? Please explain.

Response: Thank you for your question. The basis of the wear test data selected was based on the average value of the three repeated results.

Comment 7: Which international standards were tested and analyzed according to? Please specify.

Response: Thank you very much for your question. There is no international standards, the wear test and analyze.

Comment 8: Please add a figure containing the macro photograph of the test instruments and the produced materials to the material method section.

Response: Thank you very much for your valuable suggestion. The macro photograph of the test instruments and the produced materials to the material method section were as follows:

Fig Macro photograph of the (a) hot pressing sintering equipment (b) HT-1000 high temperature tribotester

Comment 9: The application of ambient temperatures in the experiment is not clearly stated. How did the authors obtain and measure temperatures?

Response: Thank you for your question. The ambient temperature were measured by thermocouple inside the furnace.

Comment 10: Write the wear rate in the material and method section, not as an expression but as an equation.

Response: Thank you very much for your valuable suggestion. The wear volume of the composite was measured by a contact probe, and the calculation formula for the volumetric wear rate is as follows:

where W is the wear rate (mm3/Nm), V is the wear volume (mm3), F is the load (N) and L is the sliding distance (m);

Comment 11: The characterization part was very short. Expand with details.

Response: Thank you very much for your valuable suggestion. The discussion of the results was enriched and the changes had been marked in the revised manuscript.

Comment 12: "We may explain this phenomenon from the following aspect: as the testing temperature in-crease from 25 °C to 250 °C, the diffusion rate of solid lubricants from matrix to sliding surface increase and moreover, the mechanical properties of PTFE matrix decrease, both of the above two situation result in the increase thickness of lubricating film, thus the COFs decrease to the minimum value [22-24]." In his statement, how did the authors come to such a conclusion when they did not examine the mechanical properties? Please explain.

Response: Thank you very much for your valuable question. In our previous published paper (Investigation of tribological characteristics of nickel alloy-based solid-lubricating composites at elevated temperatures under vacuum), results about the mechanical properties decrease with the increase of testing temperature had been reported, and it has been proved that high temperature can promote the diffusion rate of solid lubricants from matrix to sliding surface, so the author come to this conclusion.

Comment 13: The graph of COF values at different temperatures is not comparable. It can be more understandable if different graphics are created for each composite. For example, a graph for PM1 composite should include COF values at different temperatures.

Response: Thank you very much for your valuable suggestion. The graphs of COF value at different temperature for each composite were as follows.

Fig. The typical friction curve of four composites at different testing temperatures: (a) P, (b) PM1, (c) PM2, (d) PM5.

Comment 14: On what basis was the wear test data selected? Please explain.

Response: Thank you for your question. The basis of the wear test data selected was based on the average value of the three repeated results.

Comment 15: Explain the reason for the negligibility of MoS2 content at low temperatures in the paper.

Response: Thank you very much for your question. We can explain this phenomenon from the following aspects: firstly, the PTFE matrix has excellent lubricating properties, secondly, the diffusion rate of MoS2 from matrix to worn surface is low at low temperature. Combined with these two aspects, so the MoS2 content has negligibility effect on the friction and wear behavior of PTFE composite.

Comment 16: Since the experiments were repeated three times, please add the error rate to the graphs.

Response: Thank you for your valuable suggestion. The error rate were added in the graphs and the changes had been marked in the revised manuscript.

Comment 17: Discussions about the results of Raman analysis remained incomplete. For example, explain which molecular structure the peaks occur at which wavelength. Also, support the peaks with literature information.

Response: Thank you very much for your valuable suggestion. The discussions about the results of Raman analysis were enriched and the changes had been marked in the revised manuscript.

Comment 18: What is the reason for the analysis in Figure 13? Please explain the results in detail.

Response: Thank you very much for your suggestion. The reason for the analysis in Figure 13 was enriched and the changes had been marked in the revised manuscript.

Comment 19: Wear and friction do not happen on one material. The counter bodies must be analyzed (wear appearances). It would be interesting to know whether the counter body sacrifices for the body's better behavior or whether some materials were transferred from the body to the counter body or vice-versa.

Response: Thank you very much for your valuable suggestion. Indeed, by observation, there are indeed some materials were transferred from PTFE matrix to GCr15 counter body, while there is almost no GCr15 transferred to PTFE matrix.

Comment 20: SEM and EDS figures did not discuss anti-wear and anti-friction mechanisms well. Please, explain the responsible mechanisms clearly. This is one of the critical discussions this manuscript has to have in the results and discussion to show the study's novelty.

Response: Thank you very much for your valuable suggestion. The discussion about the results of SEM figures were enriched and the changes had been marked in the revised manuscript.

Comment 21: The conclusions are very weak. The findings should be restored with the correct evidence of the results obtained.

Response: Thank you very much for your valuable suggestion. The discussion of the results was enriched and the changes had been marked in the revised manuscript.

Comment 22: Please fix the typographical and eventual language problems in the paper.

Response: Thank you very much for your valuable suggestion. The typographical and eventual language problems has been carefully checked and revised throughout paper.

Round 2

Reviewer 1 Report

The manuscript has been improved and can be accepted now.

Optionally, as minor revision, the authors may add labels (a)-(d) to Figure 5 (and describe them in the caption).

The authors have to read the manuscript and polish its language one more time

Author Response

Dear reviewer,

Thank you very much for your perfect comments on our manuscript. We revised our manuscript according to your suggestions. Your comments were replied as follow.

General comments: The manuscript has been improved and can be accepted now.

Response: Thank you very much for your excellent comments.

Comment 1: Optionally, as minor revision, the authors may add labels (a)-(d) to Figure 5 (and describe them in the caption).

Response: Thank you for your suggestions, the labels (a)-(d) had been add to Figure 5 and the description about them were also presented and the variation is marked in the revised manuscript.

Reviewer 2 Report

The authors generally followed the recommendations, although there were some shortcomings. However, it is a great shortcoming that the recommendations made are not stated in the paper. It is important that the authors follow the recommendations.

·       Please indicate in the paper the recommendations mentioned in the previous revision. (For example, not using an inert gas, wear rate formula, etc.)

·       The abstract section is not well revised. Please include all significant numerical results.

·       Please explain your MoS2 weight ratio determination method in detail in the paper.

·       Usually this type of test is done according to certain standards (The Pin-on-Disk Test (DIN 50324-07, ASTM G99-05, ISO 18535). It is a major shortcoming that the experiments are not carried out according to any standard. This can cause a big trust problem for readers.

·       Please use macro photos on paper.

·       Concerns about Raman spectra not resolved. For example, explain which molecular structure the peaks occur at which wavelength. Also, support the peaks with literature information. If the authors are unable to verify the peaks obtained, this section should be deleted.

*** Authors must consider them properly before submitting the revised manuscript. A point-by-point reply is required when the revised files are submitted. In addition, revisions, as specified, should be stated in the paper. If these are not done, I do not find it appropriate to publish the paper.

Minor editing of English language required

Author Response

For reviewer 2

Dear reviewer,

Thank you very much for your carefully checks on our manuscript and giving us very valuable comments and suggestions. Responses to your comments in detail are listed below.

General comments: The authors generally followed the recommendations, although there were some shortcomings. However, it is a great shortcoming that the recommendations made are not stated in the paper. It is important that the authors follow the recommendations:

Response: Thank you very much for giving us a chance to revise our manuscript and consider the revised version.

Comment 1: Please indicate in the paper the recommendations mentioned in the previous revision. (For example, not using an inert gas, wear rate formula, etc.)

Thank you very much for your suggestions, the recommendations mentioned in the previous revision had been indicated.

Comment 2: The abstract section is not well revised. Please include all significant numerical results.

Response: Thank you very much for your suggestions, the abstract was enriched and the changes had been marked in the revised manuscript.

Comment 3: Please explain your MoS2 weight ratio determination method in detail in the paper.

Response: Thank you for your question. Firstly, the MoS2 powder was weighed with electronic balance, and then the PTFE matrix powder was also weighed, and the MoS2 weight ratio was get.

Comment 4: Usually this type of test is done according to certain standards (The Pin-on-Disk Test (DIN 50324-07, ASTM G99-05, ISO 18535). It is a major shortcoming that the experiments are not carried out according to any standard. This can cause a big trust problem for readers.

Response: Thank you very much for your valuable suggestion. Actually, it is helpful for readers when the test is done according to certain standards (DIN 50324-07, ASTM G99-05, ISO 18535). We will pay more attention to this problem in the future. In this paper, we just compared the results of four composites and investigated the influence of MoS2 content and testing temperatures (25-250 °C) on the tribological properties of PTFE composites.

Comment 5: Please use macro photos on paper.

Response: Thank you very much for your suggestions. There macro photo were as follows.

Fig. SEM images of the worn surface for P composite at different temperatures: (a) 25 °C, (b) 50 °C, (c) 200 °C, (d) 250 °C.

Fig. SEM images of the worn surface for PM1 composite at different temperatures: (a) 25 °C, (b) 50 °C, (c) 200 °C, (d) 250 °C.

Fig. SEM images of the worn surface for PM2 composite at different temperatures: (a) 25 °C, (b) 50 °C, (c) 200 °C, (d) 250 °C.

Comment 6: Concerns about Raman spectra not resolved. For example, explain which molecular structure the peaks occur at which wavelength. Also, support the peaks with literature information. If the authors are unable to verify the peaks obtained, this section should be deleted.

Response: Thank you very much for your suggestion. It have assigned the.νs(CF2) mode at 737 cm-1 in hexafluorocyclopropane.( F. A. Miller and K. O. Hartman, Spectrochim. Acta 23A, 1609 (1967).

Round 3

Reviewer 2 Report

The authors insisted they did not fulfill the issues stated in the revision. Authors should indicate the changes made in the manuscript as previously stated. Please specify the previously requested figures, formulas, and explanations in the manuscript. Otherwise, my opinion of the publication of this paper is negative.

*** Authors must consider them properly before submitting the revised manuscript. A point-by-point reply is required when the revised files are submitted. In addition, revisions, as specified, should be stated in the paper. If these are not done, I do not find it appropriate to publish the paper.

Minor editing of English language required

Author Response

For reviewer 2

Dear reviewer,

Thank you very much for your carefully checks on our manuscript and giving us very valuable comments and suggestions. Responses to your comments in detail are listed below.

General comments: The authors insisted they did not fulfill the issues stated in the revision. Authors should indicate the changes made in the manuscript as previously stated. Please specify the previously requested figures, formulas, and explanations in the manuscript. Otherwise, my opinion of the publication of this paper is negative:

Response: Thank you very much for giving us a chance to revise our manuscript and consider the revised version. According to your suggestions, the authors had been added all the previously requested figures, formulas, and explanations in the manuscript, and the changes made in the manuscript as previously stated had been indicated.
